# ENERGY CONSUMPTION-AWARE TABULAR BENCHMARKS FOR NEURAL ARCHITECTURE SEARCH

## ABSTRACT

The demand for large-scale computational resources for Neural Architecture Search (NAS) has been lessened by tabular benchmarks for NAS. Evaluating NAS strategies is now possible on extensive search spaces and at a moderate computational cost. But so far, NAS has mainly focused on maximising performance on some hold-out validation/test set. However, energy consumption is a partially conflicting objective that should not be neglected. We hypothesise that constraining NAS to include the energy consumption of training the models could reveal a subspace of undiscovered architectures that are more computationally efficient with a smaller carbon footprint. To support the hypothesis, an existing tabular benchmark for NAS is augmented with the energy consumption of each architecture. We then perform multi-objective optimisation that includes energy consumption as an additional objective. We demonstrate the usefulness of multi-objective NAS for uncovering the trade-off between performance and energy consumption as well as for finding more energy-efficient architectures. The updated tabular benchmark, `EC-NAS-Bench`, is open-sourced to encourage the further exploration of energy consumption-aware NAS.

## 1 INTRODUCTION

The design of neural architectures is a complex task. While general guidelines for producing *suitable* neural architectures have been proposed, neural architecture design still requires expert domain knowledge, experience, and not least substantial effort (Philipp, 2021; Zoph & Le, 2016; Ren et al., 2020). This led to an upsurge in research on automated exploration and design of neural architectures cast as an optimisation problem – neural architecture search (NAS) (Baker et al., 2016; Zoph & Le, 2016; Real et al., 2017).

NAS strategies explore neural architectures in a predefined search space relying on model training and evaluation to determine the model's fitness (i.e., validation/test set score) to adjust the search strategy and extract the best performing architecture (Ren et al., 2020). NAS strategies have shown great promise in discovering novel architecture designs yielding state-of-the-art model performance (Liu et al., 2017; 2018; Lin et al., 2021; Baker et al., 2017). However, it can be prohibitively expensive to perform NAS (Tan & Le, 2019b) due to the demand for large-scale computational resources and the associated carbon footprint of NAS (Schwartz et al., 2019; Anthony et al., 2020).

The introduction of tabular benchmarks for NAS significantly lessened the computational challenges mentioned above by facilitating the evaluation of NAS strategies on a limited search space of architectures (Klein & Hutter, 2019; Dong & Yang, 2020). Predictive models and zero- and one-shot models (Wen et al., 2019; Lin et al., 2021; Zela et al., 2020) have reduced time-consuming model training and thereby increased the efficiency of NAS strategies. Most recently, surrogate NAS benchmarks (Zela et al., 2022) have been proposed for arbitrary expansion of architecture search spaces for NAS.

Notwithstanding the aforementioned major contributions to the advancement of NAS research, the prime objective of NAS has been maximising a performance objective on some hold-out test/validation test. NAS strategies can be evaluated effectively, yet the search strategies do not intentionally aim to find computationally efficient architectures. That is, the NAS may efficiently determine model performance at a moderate computational cost, but energy efficiency is generally not an objective of NAS.

We hypothesise that adding the energy consumption of training models as a NAS objective could reveal a sub-space of computationally efficient models that also have a smaller carbon footprint. In order to find efficient architectures without sacrificing cardinal performance requirements, we propose the use of NAS strategies that will optimise for multiple objectives.

**Our main contributions.**

1. We provide an energy consumption-aware tabular benchmark for NAS based on NAS-Bench-101 (Ying et al., 2019). For each architecture, we added its training energy consumption, power consumption and carbon footprint. We hope that the new data set will foster the development of environmentally friendly deep learning systems.

2. We also introduce a surrogate energy model to predict the training energy cost for a given architecture in a large search space (about $423k$ architectures)

3. To exemplify the use of the new benchmark, we devise a simple multi-objective optimisation algorithm for NAS and apply it to optimise generalisation accuracy as well as energy consumption.

4. We demonstrate the usefulness of multi-objective architecture exploration for revealing the trade-off between performance and energy efficiency and for finding efficient architectures obeying accuracy constraints. This is also demonstrated with other baseline multi-objective methods.

## 2 ENERGY CONSUMPTION-AWARE BENCHMARKS - `EC-NAS-Bench`

Our energy consumption-aware tabular benchmark `EC-NAS-Bench` is based on Nas-Bench-101 (Ying et al., 2019). We closely follow their specification of architectures; however, the search space of architectures that are considered, the evaluation approach and the metrics provided for each architecture is different. This section will briefly present `EC-NAS-Bench` and its differences to NAS-Bench-101.

### 2.1 ARCHITECTURE DESIGN

**Network Topology.** All architectures considered are convolutional neural networks (CNNs) designed for the task of image classification on CIFAR-10 (Krizhevsky, 2009). Each neural network comprises a convolutional stem layer followed by three repeats of three stacked *cells* and a downsampling layer. Finally, a global pooling layer and a dense softmax layer are used. The space of architectures, $\mathbb{X}$, is limited to the topological space of *cells*, where each cell is a configurable feedforward network.

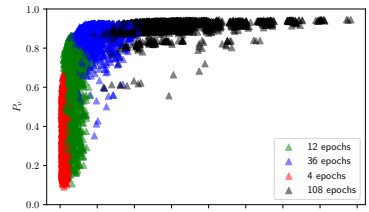

Figure 1: Visualisation of the 5V space of architectures with the validation performance $P_v$ and the corresponding training energy cost $(E)$ for four training epoch budgets (4,12,36,108).

**Cell Encoding.** The individual *cells* are represented as directed acyclic graphs (DAGs). Each DAG, $G(V, M)$, has $N = |V|$ vertices (or nodes) and edges described in the binary adjacency matrix $M \in \{0, 1\}^{N \times N}$. The set of operations (labels) that each node can realise is given by $\mathcal{L}' = \{\texttt{input}, \texttt{output}\} \cup \mathcal{L}$, where $\mathcal{L} = \{\texttt{3x3conv}, \texttt{1x1conv}, \texttt{3x3maxpool}\}$. Two of the $N$ nodes are always fixed as `input` and `output` to the network. The remaining $N - 2$ nodes can take up one of the labels in $\mathcal{L}$. The connections between nodes of the DAG are encoded in the upper-triangular adjacency matrix with no self-connections (zero main diagonal entries). For a given architecture, $\mathcal{A}$, every entry $\alpha_{i,j} \in M_{\mathcal{A}}$ denotes an edge, from node $i$ to node $j$ with operations $i, j \in \mathcal{L}$ and its labelled adjacency matrix, $L_{\mathcal{A}} \in M_{\mathcal{A}} \times \mathcal{L}'$.

**Search space.** The number of DAGs grows exponentially with $N$ and $L$ (Ying et al., 2019). We restrict the search space in `EC-NAS-Bench` by imposing $|V| \leq 5$ and $|A \neq 0| \leq 9$, referred to as the 5V space. The search space with $|V| \leq 4$ called 4V space is also considered. In contrast, NAS-Bench-101 considers the search space for $|V| \leq 7$. With these imposed restrictions on the

search space of `EC-NAS-Bench`, 91, 2532 and $423k$ unique architectures are identified from the 4V, 5V and 7v spaces, respectively.

## 2.2 ENERGY CONSUMPTION-AWARENESS

Resource-constrained NAS for obtaining efficient architectures has been explored mainly by optimising the total number of floating point operations (FPOs) (Tan & Le, 2019a). Optimising for FPOs, however, might not be entirely indicative of the efficiency of models (Henderson et al., 2020). It has been reported that models with fewer FPOs have bottleneck operations that can consume the bulk of the training time (Howard et al., 2017), and some models with high FPOs have lower inference time (Jeon & Kim, 2018). Energy consumption optimised hyperparameter selection outside of NAS settings for large language models has been recently investigated in Puvis de Chavannes et al. (2021).

The energy consumption during the training of a model encapsulates facets of architecture efficiency that are not entirely taken into consideration when using standard resource constraints such as FPOs, computational time and the number of parameters. Energy consumption accounts for both hardware and software variations in the experimental set-ups. To foster a new direction for NAS to find more efficient architectures, we use energy consumption as the additional objective along with standard performance measures.

## 2.3 QUANTIFYING ENERGY CONSUMPTION

About 75% of the total energy costs during training a neural network are incurred by hardware accelerators such as graphics processing units (GPUs) or tensor processing units (TPUs) (Dodge et al., 2022). The remaining energy consumption is mainly due to the central processing units (CPUs) and dynamic random access memory (DRAM). Additional energy consumed by the supporting infrastructure, such as cooling- and power systems and dissipation, is usually accounted for by the power usage effectiveness (PUE), which is an overhead factor. Several open-source tools have been published in the past couple of years, such as *experiment-impact-tracker* (Henderson et al., 2020), *Carbontracker* (Anthony et al., 2020) and *CodeCarbon* (Schmidt et al., 2021) provide convenient ways to track and log the energy consumption of neural networks by taking these factors into consideration.

In `EC-NAS-Bench`, the energy consumption of training and evaluating the neural architectures is estimated by modifying the tool Carbontracker (Anthony et al., 2020). Our version of the tool monitors the GPUs, CPUs and DRAM and estimates the total energy costs, $E$ (kWh), aggregate carbon footprint (kgCO$_2$eq) based on the instantaneous carbon intensity of the regions and the total computation time, $T$(s). The complete set of metrics that are measured and reported in `EC-NAS-Bench` are listed in Table 1.

## 2.4 ARCHITECTURE PERFORMANCE AND EFFICIENCY

**Training Pipeline.** Architectures from the 4V and 5V space are trained on CIFAR-10 (Krizhevsky, 2009) using 40k samples and evaluated on 10k validation and test samples (60k total). Each model is trained on an in-house Slurm cluster on a single NVIDIA Quadro RTX 6000 GPU with 24 GB memory and two Intel CPUs. The training strategy, or hyper-parameter setting, is similar to that of NAS-Bench-101 (Klein & Hutter, 2019). Pre-

| Metrics | Unit of measurement | Notation |
|---|---|---|
| Model parameters | Million (M) | $\|\theta\|$ |
| Test/Train/Eval. time | Seconds (s) | $T(s)$ |
| Test/Train/Val. Acc. | $\mathbb{R} \in [0; 1]$ | $P_v$ |
| Energy consumption | Kilowatt-hour (kWh) | $E$(kWh) |
| Power consumption | Joule (J), Watt (W) | $E$(J), $E$(W) |
| Carbon footprint | kgCO$_2$eq | – |
| Carbon intensity | g/kWh | – |

Table 1: Metrics reported in `EC-NAS-Bench`.

dicting the energy consumption of longer model runs from a few training epochs has been shown to be robust when performed on the same hardware (Anthony et al., 2020). To refrain from re-training and re-evaluating all the models in NAS-Bench-101, we train each model for only 4 epochs and then obtain surrogate time and energy measurements by linear scaling. We then tabulate these measurements along with the corresponding mean performance metrics for each model from NAS-Bench-101 and obtain metrics for training and evaluating each model for 12, 36 and 108 epochs.

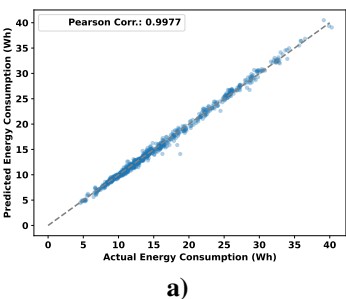 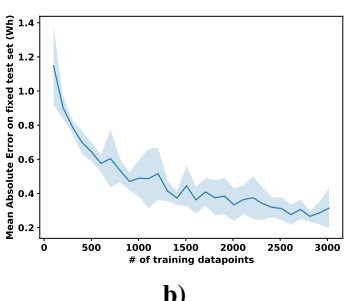

**a)**             **b)**

Figure 2: a) Scatter plot showing the correlation between the predicted and actual energy consumption for a subset of the architectures from the 7V-space. b) Influence of the number of training data points on the test set performance in the Surrogate 7V-space model.

**Metrics.** We report the operations, no. parameters, and performance metrics in `EC-NAS-Bench`, as in NAS-Bench-101, and additionally, we include efficiency measures in terms of energy consumption and the carbon footprint for training each model. The primary focus for efficiency metrics is to quantify the resource costs specific to model training; however, we also report the total resource costs, which include computational overhead, e.g., data movements. For completeness, we also provide carbon intensity measures at training time, timestamp, and average energy consumption of computing resources. We have made the metrics of each architecture readily accessible to encourage the development of NAS strategies for exploring efficient architectures. The metrics reported relevant to this work can be seen in Table 1.

## 2.5 SURROGATE DATASET FOR 7V-SPACE

The 4V and 5V search spaces are the primary spaces used in this work to reduce the overall resource consumption to populate the energy measurements in the tabular benchmark datasets. However, even the 5V space has only a fraction of possible architectures compared to the 7V space published in Ying et al. (2019), which has about $423k$ architectures. Computing the energy consumption as done for 4V and 5V datasets on the 7V space is prohibitively expensive[1].

We instead sample a subset of architectures from the 7V space and obtain the actual energy costs for $4300$ architectures. Using these measurements we train a multi-layered perceptron (MLP) based surrogate energy prediction model. The MLP takes the graph-encoded architecture and the number of parameters as input and predicts the energy consumption for a given number of epochs. This surrogate model is similar to recent surrogate NAS methods that have shown to be more efficient Zela et al. (2022). Details of the surrogate model used to predict the energy measurements for the 7V space are provided Appendix D.

The resulting surrogate 7V dataset with the energy measurements yields a close approximation of the actual training energy costs as shown in Figure 2-a). The Pearson correlation between the actual and predicted energy measurements is $0.9977$. In Figure 2-b), we also show that the mean absolute error of the predicted- and actual energy measurements plateau with about $3000$ architectures, justifying its use to predict on the remaining 7V space. The standard deviation is estimated over 10 random initialisations of the surrogate model per training dataset size.

## 2.6 INFLUENCE OF HARDWARE ON `EC-NAS-Bench`

The energy consumption of the architectures in the 4V and 5V spaces were obtained on a single RTX Quadro 6000 GPU. While the energy measurements tabulated in `EC-NAS-Bench` are specific to these hardware settings, we argue that the trends across the architectures hold independent of the actual hardware used. To demonstrate this, we trained the architectures in the 4V space on four different (Nvidia) GPUs spanning multiple generations: Titan XP, RTX 3060, RTX 3090 and RTX Quadro 6000.

While the energy consumed by each model on specific hardware is different, the trends compared to other models are maintained across different GPUs. This is captured in Figure 3, where the energy

---

[1]Our estimates showed that it would require 770 GPU days of compute.

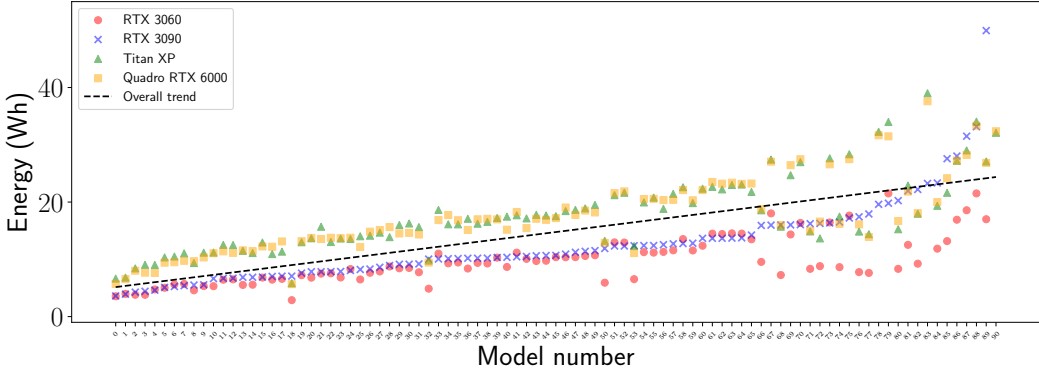

Figure 3: Energy consumption of all 4V models on four different GPUs. The models are sorted based on their energy consumption for improved visualisation.

consumption for each architecture in the 4V space on all four GPUs is reported. This trend confirms the fact that when NAS is constrained on energy consumption and performance the resulting models would remain the same irrespective of the specific hardware used.

## 3 NAS STRATEGIES WITH `EC-NAS-Bench`

Given a tabular benchmark which can be used to query for model training energy consumption in addition to other standard metrics such as in `EC-NAS-Bench`, NAS strategies can be used to search for energy-efficient architectures. We next present multi-objective optimisation as a suitable strategy to uncover the trade-off between performance and efficiency, which supports an energy-aware architecture choice.

### 3.1 MULTI-OBJECTIVE OPTIMISATION

Multi-objective optimisation (MOO) simultaneously optimises several, potentially conflicting objectives. The goal of MOO is to find or to approximate the set of Pareto-optimal solutions, where a solution is Pareto-optimal if it cannot be improved in one objective without getting worse in another.

In this work, we introduce a simple evolutionary MOO algorithm (SEMOA) based on Krause et al. (2016). The algorithm is simple, but derived from canonical principles of derivative-free multi-criteria optimisation, such as hypervolume maximisation. Details of SEMOA are presented in Appendix A.2. We also use several existing MOO algorithms: random search, Speeding up Evolutionary Multi-Objective Algorithms (SHEMOA) and Mixed Surrogate Expected Hypervolume Improvement (MSEHVI), implemented in Izquierdo et al. (2021) to demonstrate the usefulness of `EC-NAS-Bench`.

### 3.2 EVALUATION OF NAS STRATEGIES

**Experimental Setup.** We conduct experiments on `EC-NAS-Bench` by adapting the presented MOO-algorithm to perform both single-objective optimisation (SOO) and MOO. In the former, we will naturally find only one solution when optimising a single objective. In contrast, when optimising multiple, diverse objectives, we will find the empirical Pareto-front in the latter. We run the algorithm in the 4V and 5V space of models trained for 108 epochs. The optimisation is performed over 100 evolutions with a population size of 20. All the experiments are conducted on a desktop workstation with a single NVIDIA RTX 3090 GPU with 24GB memory and Intel(R) Core(TM) i7-10700F CPU @ 2.90GHz.

**Performance Criteria.** For the multi-objective optimisation, we use the validation accuracy ($P_v$) and the training energy cost, $E$(kWh), as the two objectives to be jointly optimised using the MOO algorithm. For the single-objective optimisation, we only use $P_v$ as the performance objective. We

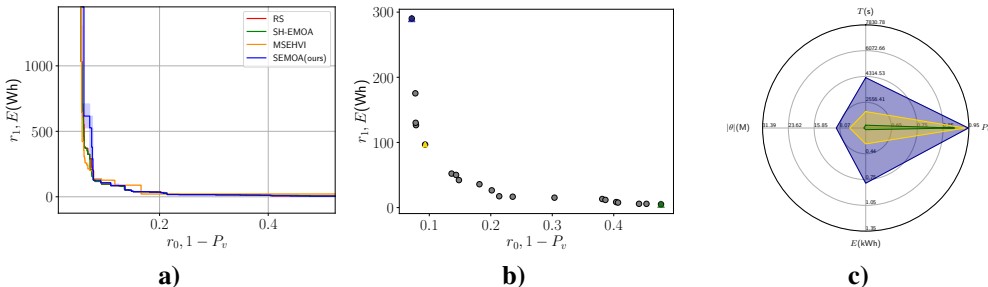

Figure 4: Multi-objective exploration of 5V space. **a)** The attainment curve showing the median solutions and the inter-quartile variations for 100 random initialisations of the four MOO algorithms on the `EC-NAS-Bench` dataset. The two objectives being optimised $(1 - P_v, E)$ are shown on the two axes. **b)** Pareto front obtained for one of the MOO runs shows the family of solutions as discrete points **c)** Radar plot showing three solutions from a Pareto front that is close to the median in subfigure a). The three architectures are chosen to correspond to the two extrema ($\mathcal{A}_{\mathbf{r}_0}$ in blue, $\mathcal{A}_{\mathbf{r}_1}$ in green) and the knee point ($\mathcal{A}_{\mathbf{r}_k}$ in yellow). The plot has four axes capturing the performance $P_v$ and resource consumption measured in $E, T, |\theta|$.

use energy cost rather than, e.g., training time, considering that $E$ is agnostic to parallel computing. We note that it is possible to use any of the provided metrics in Table 1 for the purpose of single- and multi-objective optimisation. As the MOO algorithm minimises the objectives, we simply use the negative of the objectives in cases where the quantities are to be maximised; for instance, we optimise $-P_v$ as accuracy is a maximisation objective.

**Training costs** In aggregate, `EC-NAS-Bench` had a total estimated training cost of 124.214 GPU days, 2021.02 kWh and 259.047 kgCO$_2$eq for the 5V space. The 4V space had a total estimated training cost of 3.854 GPU days, 63.792 kWh and 5.981 kgCO$_2$eq. The actual training costs for the 5V space were only 3.105 GPU days, 50.525 kWh and 6.476 kgCO$_2$eq. Actual training costs of the 4V space were 0.096 GPU days, 1.594 kWh and 0.149 kgCO$_2$eq.

In total, we saved an estimated compute cost of 121.109 GPU days, 1970.495 kWh and 252.571 kgCO$_2$eq for the 5V space, and 3.758 GPU days, 48.931 kWh and 6.327 kgCO$_2$eq for the 4V space. We obtain $\approx 97\%$ reduction in computing resources and energy consumption in all efficiency measures.

## 4 RESULTS

**Multi-objective exploration of 5V space.** The key results from the experiments on `EC-NAS-Bench` using the multi-objective optimisation of $E$ and $-P_v$ are shown in Figure 4-a),b) and c). Pareto fronts over multiple random initialisations of the four MOO algorithms: SEMOA (ours), Random Search, SHEMOA, MSEHVI, are visualised as attainment curves in Figure 4-a) which summarises the median solutions attained over the multiple runs(Fonseca et al., 2001). All the MOO algorithms are able to explore the search space reasonably well, yielding attainment curves that largely look similar.

The Pareto front obtained from the our MOO algorithm, SEMOA, for one run is shown in Figure 4-b). It also shows the extrema $(\mathbf{r}_0, \mathbf{r}_1)$ on both ends of the front preferring one of the objectives, whereas the knee point $(\mathbf{r}_k)$ offers the best trade-off between the two objectives. These three points are shown in different colours and markers, where the two extrema ($\mathcal{A}_{\mathbf{r}_0}$/Blue, $\mathcal{A}_{\mathbf{r}_1}$/Green) and the knee point ($\mathcal{A}_{\mathbf{r}_k}$/yellow). We compute the bend-angles to find the knee point as suggested by Deb & Gupta (2011).

The architectures corresponding to the two extrema ($\mathcal{A}_{\mathbf{r}_0}, \mathcal{A}_{\mathbf{r}_1}$) and the corresponding knee point ($\mathcal{A}_{\mathbf{r}_k}$) for a single MOO run are visualised in the radar plot in Figure 4-b). The exact performance metrics for the three models in Figure 4-b) are also reported in Table 2. The solution covering the largest area is one of the extremal points ($\mathcal{A}_{\mathbf{r}_0}$, blue) with high accuracy (0.944) but also a larger footprint in the energy consumption (1.032kWh), computation time (5482.78s) and the number of

| Model | Strategy | $|\theta|$(M)↓ | $T(s)$ ↓ | $P_v$ ↑ | $E$(kWh)↓ |
|---|---|---|---|---|---|
| $\mathcal{A}_{\mathbf{r}_0}$ (B) | SEMOA | 12.08 | 3878.66 | 0.942 | 0.709 |
| $\mathcal{A}_{\mathbf{r}_1}$ (G) | SEMOA | **0.88** | **818.94** | 0.834 | **0.137** |
| $\mathcal{A}_{\mathbf{r}_k}$ (Y) | SEMOA | 5.09 | 1916.65 | 0.932 | 0.324 |
| $\mathcal{A}_{\mathbf{r}*}$ (R) | SOO | 21.22 | 5482.78 | **0.944** | 1.032 |

Table 2: Metrics for 5V space models in single- and multi-objective setting seen in Figure 4. For SOO the optimal solution ($\mathcal{A}_{\mathbf{r}*}$/Red) is reported. For MOO the two extrema ($\mathcal{A}_{\mathbf{r}_0}$/Blue, $\mathcal{A}_{\mathbf{r}_1}$/Green) and the knee point ($\mathcal{A}_{\mathbf{r}_k}$/yellow) are reported.

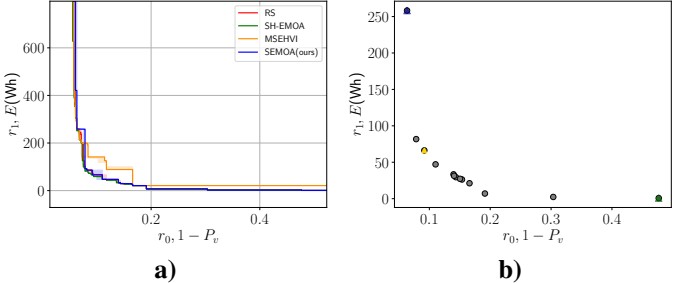

**a)**            **b)**

Figure 5: Multi-objective exploration of 7V space. **a)** The attainment curve showing the median solutions and the inter-quartile variations for 100 random initialisations of the four MOO algorithms on the EC-NAS-Bench dataset. The two objectives being optimised $(1 - P_v, E)$ are shown on the two axes. **b)** Pareto front obtained for one of the MOO runs shows the family of solutions as discrete points

parameters (21.22M) compared to the other extremum ($\mathcal{A}_{\mathbf{r}_1}$, green) or the knee point ($\mathcal{A}_{\mathbf{r}_k}$, yellow). The model corresponding to the knee point ($\mathcal{A}_{\mathbf{r}_k}$) provides a large reduction in the energy consumption (0.324kWh) at the expense of a small reduction in performance (0.932).

**Single-objective exploration.** We optimise only the validation accuracy, $P_v$, to simulate standard NAS practices. The resulting solution is shown the last row of Table 2. This SOO model achieves the highest validation accuracy (0.944). However, the footprint of the solution along the energy consumption, computation time and the number of parameter axes are larger than those from the MOO algorithm.

**Multi-objective exploration of 7V space.** The MOO results for the surrogate 7V space resemble the trends observed in the 5V space, as shown in Figure 5. As with the 5V space all the attainment curves of all the four MOO algorithms look similar. Visibly, the MSEHVI method seems to underperform compared to other models due to the protrusion around the knee-point compared to the other models, which are largely overlapping. A single Pareto front of SEMOA are also showed in Figure 5-b), with trends comparable with those in the 5V space results in Figure 4.

## 5 DISCUSSIONS

**Single versus multi-objective optimisation.** The performance trends of the SOO and MOO solutions are clearly captured in Table 2. The knee point solution, $\mathcal{A}_{\mathbf{r}_k}$, from MOO, yields an architecture that consumes about 70% less energy and has only about 1% degradation in performance. Depending on the downstream tasks, this could be a reasonable trade-off. If the degradation in performance cannot be tolerated, the Pareto front offers other candidate solutions for the practitioners to choose from. For instance, the extremum solution ($\mathcal{A}_{\mathbf{r}_0}$) offers basically the same performance as the SOO solution by consuming about 32% less energy.

**Training time is not an alternative to energy consumption.** The original NAS-Bench-101 already reports the training time (Ying et al., 2019). In single hardware regimes, this could serve as a measure of the energy consumption, as training time mostly correlates with the energy consumption. However, as most neural architecture training is performed on multiple GPUs with large-scale parallelism, training time alone cannot capture the efficiency of models. Aggregate energy consumption can take parallel hardware and the associated overheads into consideration. Even in single GPU

training settings, energy consumption could optimise for energy-efficient models. For instance, a small architecture trained on a large GPU still has larger energy consumption due to the under-utilisation of the hardware resources. In such instances, a larger model could (to a certain extent) yield more performance improvements for the total energy consumed ($P_v/E$).

**Energy efficient tabular NAS benchmark for obtaining efficient architectures.** Tabular benchmarks such as NAS-Bench-101 (Ying et al., 2019) were introduced to reduce the resources required to perform NAS. However, even the one-time cost of generating a tabular benchmark dataset is massive. Surrogate NAS benchmarks are being studied to alleviate these costs, where the models are not exhaustively trained and evaluated. Instead, the performance metrics of architectures are estimated based on smaller training costs. For instance, this is achieved using predictive modelling based on learning curves (Yan et al., 2021), gradient approximations (Xu et al., 2021), or by fitting surrogate models to a subset of architectures (Zela et al., 2022). Similar to these attempts, the proposed `EC-NAS-Bench` dataset does not train all the models but bases its predictions on training the models only for 4 epochs, as described in Section 3.2. This results in about 97% reduction if the dataset were to be created from scratch, as shown in Table 3. Thus, `EC-NAS-Bench` is an energy-efficient tabular benchmark that can be used to obtain energy-efficient architectures as demonstrated in Section 4.

**Carbon-footprint aware NAS.** The `EC-NAS-Bench` dataset reports several metrics per architecture, as shown in Table 1. Combinations of these metrics and the use of MOO could allow for the exploration of architecture spaces that have interesting properties. For instance, NAS can be performed to directly optimise the carbon footprint of deep learning models. Although instantaneous energy consumption and carbon footprint are

| Space | Red. GPU days | Red. kWh | Red. kgCO$_2$eq |
|-------|---------------|----------|-----------------|
| 4V    | 3.758         | 48.931   | 6.327           |
| 5V    | 121.109       | 1970.495 | 252.571         |

Table 3: Estimated reduction in total training costs w.r.t GPU days, kWh and kgCO$_2$eq of the 4V and 5V space.

linearly correlated, when measured over a longer duration (>5m) these quantities differ due to the fluctuations of the instantaneous carbon intensity (Anthony et al., 2020). These carbon intensity fluctuations are caused by the variations of the power sources to the grid (Henderson et al., 2020). This can have implications when training models for a longer duration or on cloud instances that can be distributed over data centres in different countries (Dodge et al., 2022). By reporting instantaneous and aggregate carbon footprint of model training in `EC-NAS-Bench` we facilitate the possibility of carbon footprint aware NAS (Selvan et al., 2022). In this work, we focused only on energy consumption awareness to work around the temporal- and spatial variations of the carbon intensity.

**Energy Consumption aware Few-shot NAS.** Tabular benchmarks such as NAS-Bench-101 (Ying et al., 2019) provide an efficient way to explore different NAS strategies where the model training cost is only incurred once. One restriction with such tabular benchmarks is that they are specific to a set of architectures (fx: feedforward convolutional neural networks) and datasets (fx: CIFAR-10). Developing tabular benchmarks for all possible network architectures and datasets is alleviated using one- or few- shot learning methods (Zhao et al., 2021; Zela et al., 2020). Integrating surrogate models for predicting learning dynamics (Zela et al., 2022) and energy measurements using the surrogate model in Section 2.5 could bridge the divide between few-shot and surrogate tabular benchmark datasets that are also energy consumption-aware. We have demonstrated the integration of surrogate energy models with existing tabular benchmark datasets, and extending these to surrogate benchmark datasets is straightforward.

**Limitations.** Constraining the number of vertices in the DAGs results in sparser search spaces for the optimisation strategy. The optimisation strategy will therefore be more sensitive to initialisation and choice of random seeds, and the empirical Pareto front will appear to be more rigid, as seen in the attainment plot in Figure 4-c, even when multiple initialisations and trials are carried out. We also only demonstrated experiments on the 4V and 5V spaces.

To reduce the computation cost, in `EC-NAS-Bench` we used the surrogate time and energy measurements that do not model training time variability. We also query the performance metrics from the three repeats of NAS-Bench-101 and update `EC-NAS-Bench` with their mean performance metrics.

All these limitations are primarily driven by the need to minimise the energy consumption of these experiments. While these are at the expense of variability, we argue that the resulting reduction in the energy consumption justifies these choices. Further, the results from these small-scale experiments have been shown to extend to larger space of architectures (Ying et al., 2019).

## 6 CONCLUSIONS AND FUTURE WORK

In this work, we presented an updated tabular benchmark dataset, `EC-NAS-Bench`, which tabulates the energy consumption and carbon footprint of training models, in addition to standard performance measures. Using multi-objective optimisation strategies, we showed that Pareto-optimal solutions offer appealing trade-offs between the performance measures and the energy consumption of model training. We qualitatively showed that large reductions (about 70%) in energy consumption are possible with <1% reduction in performance.

In addition to providing energy consumption measures, the `EC-NAS-Bench` benchmark provides metrics such as average carbon footprint and power consumption of CPUs, GPUs and DRAM. We hope this will foster interest in the development of models that are efficient and environmentally friendly by optimising for their energy consumption and carbon footprint.

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

# A  MULTI-OBJECTIVE OPTIMISATION

Formally, let the MOO problem be described by $\boldsymbol{f} \colon \mathbb{X} \to \mathbb{R}^m$, $\boldsymbol{f}(x) \mapsto (f_1(x), \ldots, f_m(x))$. Here $\mathbb{X}$ denotes the search space of the optimization problem and $m$ refers to the number of objectives. We assume w.l.o.g. that all objectives are to be minimized. For two points $x, x' \in \mathbb{X}$ we say that $x'$ *dominates* $x$ and write $x' \prec x$ if $\forall i \in \{1, \ldots, m\} \colon f_i(x') \leq f_i(x) \land \exists j \in \{1, \ldots, m\} \colon f_j(x') < f_j(x)$. For $X', X'' \subseteq \mathbb{X}$ we say that $X'$ dominates $X''$ and write $X' \prec X''$ if $\forall x'' \in X'' \colon \exists x' \in X' \colon x' \prec X''$. The subset of non-dominated solutions in a set $X' \subseteq \mathbb{X}$ is given by $\mathrm{ndom}(X') = \{x \mid x \in X' \land \nexists x' \in X' \setminus \{x\} \colon x' \prec x\}$. The *Pareto front* of a set $X' \subset \mathbb{X}$ defined as $\mathcal{F}(X') = \{\boldsymbol{f}(x) \mid x \in \mathrm{ndom}(X')\}$ and, thus, the goal of MOO can be formalised as approximating $\mathcal{F}(\mathbb{X})$.

In iterative MOO, the strategy is to step-wise improve a set of candidate solutions towards a sufficiently good approximation of $\mathcal{F}(\mathbb{X})$. For the design of a MOO algorithm, it is important to have a way to rank two sets $X'$ and $X''$ w.r.t. the overall MOO goal even if neither $X' \prec X''$ nor $X'' \prec X'$. This ranking can be done by the hypervolume measure. The hypervolume measure or $\mathcal{S}$-metric (see Zitzler & Thiele, 1999) of a set $X' \subseteq \mathbb{X}$ is the volume of the union of regions in $\mathbb{R}^m$ that are dominated by $X'$ and bounded by some appropriately chosen reference point $\boldsymbol{r} \in \mathbb{R}^m$:

$$\mathcal{S}_{\boldsymbol{r}}(X') := \Lambda \left( \bigcup_{x \in X'} \left[ f_1(x), r_1 \right] \times \cdots \times \left[ f_m(x), r_m \right] \right),$$

where $\Lambda(\cdot)$ is the Lebesgue measure. The hypervolume is, up to weighting objectives, the only strictly Pareto compliant measure (Zitzler et al., 2003) in the sense that given two sets $X'$ and $X''$ we have $\mathcal{S}(X') > \mathcal{S}(X'')$ if $X'$ dominates $X''$. As stated by Bringmann et al. (2013), the worst-case approximation factor of a Pareto front $\mathcal{F}(X')$ obtained from any hypervolume-optimal set $X'$ with size $|X'| = \mu$ is asymptotically equal to the best worst-case approximation factor achievable by any set of size $\mu$, namely $\Theta(1/\mu)$ for additive approximation and $1 + \Theta(1/\mu)$ for relative approximation (Bringmann & Friedrich, 2013). Now we define the *contributing hypervolume* of an individual $x \in X'$ as

$$\Delta_{\boldsymbol{r}}(x, X') := \mathcal{S}_{\boldsymbol{r}}(X') - \mathcal{S}_{\boldsymbol{r}}(X' \setminus \{x\}) .$$

The value $\Delta(x, X')$ quantifies how much a candidate solution $x$ contributed to the total hypervolume of $X'$ and can be regarded as a measure of the relevance of the point. Therefore, the contributing hypervolume is a popular criterion in MOO algorithms (e.g. Beume et al., 2007; Igel et al., 2007; Bader & Zitzler, 2011; Krause et al., 2016). If we iteratively optimize some solution set $P$, then points $x$ with low $\Delta(x, P)$ are candidates in an already crowded region of the current Pareto front $\mathcal{F}(P)$, while points with high $\Delta(x, P)$ mark areas that are promising to explore further.

## A.1  SEMOA: SIMPLE EVOLUTIONARY MULTI-OBJECTIVE OPTIMISATION ALGORITHM

In this study, we used a simple MOO algorithm based on hypervolume maximisation outlined in Algorithm 1 inspired by Krause et al. (2016). The algorithm iteratively updates a set $P$ of candidate solutions, starting from a set of random network architectures. Dominated solutions are removed from $P$. Then $\lambda$ new architectures are generated by first selecting $\lambda$ architectures from $P$ and then modifying these architectures according to the perturbation described in Procedure 2. The $\lambda$ new architectures are added to $P$ and the next iteration starts. In Procedure 2, the probability $p_{\mathrm{edge}}$ for changing (i.e., either adding or removing) an edge is chosen such that in expectation, two edges are changed, and the probability $p_{\mathrm{node}}$ for changing a node is set such that in expectation every second perturbation changes the label of a node.

The selection of the $\lambda > m$ architectures from the current solution set is described in Procedure 3. We always select the *extreme points* in $P$ that minimize a single objective (thus, the precise choice of the reference point $\boldsymbol{r}$ is of lesser importance). The other $m - \lambda$ points are randomly chosen preferring points with higher contributing hypervolume. The points in $P$ are ranked according to their hypervolume contribution. The probability of being selected depends linearly on the rank. We use *linear ranking selection* (Baker, 1985; Greffenstette & Baker, 1989), where the parameter controlling the slope is set to $\eta^+ = 2$. Always selecting the extreme points and focusing on points with large contributing hypervolume leads to a wide spread of non-dominated solutions.

---

**Algorithm 1** SEMOA for NAS strategy

---

**Input:** objective $\boldsymbol{f} = (f_1, \ldots, f_m)$, maximum number of iterations $n$
**Output:** set of non-dominated solutions $P$

1: Initialize $P \subset \mathbb{X}$ (e.g., randomly)                      $\triangleright$ Initial random architectures
2: $P \leftarrow \mathrm{ndom}(P)$                                          $\triangleright$ Discard dominated solutions
3: **for** $i \leftarrow 1$ to $n$ **do**                                    $\triangleright$ Loop over iterations
4:     $O \leftarrow \mathrm{LinearRankSample}(P, \lambda)$                  $\triangleright$ Get $\lambda$ points from $P$
5:     $O \leftarrow \mathrm{Perturb}(O)$                                    $\triangleright$ Change the architectures
6:     Compute $\boldsymbol{f}(x)$ for all $x \in O$                         $\triangleright$ Evaluate architectures
7:     $P \leftarrow \mathrm{ndom}(P \cup O)$                                $\triangleright$ Discard dominated points
8: **end for**
9: **return** $P$

---

**Procedure 2** Perturb($O$)

---

**Input:** set of architectures $O$, variation probabilities for edges and nodes $p_{\mathrm{edge}}$ and $p_{\mathrm{node}}$
**Output:** set of modified architecture $O^*$

1: **for all** $M_{\mathcal{A}} \in O$ **do**                               $\triangleright$ Loop over matrices
2:     **repeat**
3:         **for all** $\alpha_{i,j} \in M_{\mathcal{A}}$ **do**             $\triangleright$ Loop over entries
4:             With probability $p_{\mathrm{edge}}$ flip $\alpha_{i,j}$
5:         **end for**
6:         **for all** $l \in L_{\mathcal{A}}$ **do**                        $\triangleright$ Loop over labels
7:             With probability $p_{\mathrm{node}}$ change the label of $l$
8:         **end for**
9:     **until** $M_{\mathcal{A}}$ has changed
10: **end for**
11: **return** $O^*$

---

### A.2 MULTI-OBJECTIVE OPTIMISATION BASELINES

**Hyperparameters for the MOO baseline methods** All baseline methods utilise the tabular benchmarks of `EC-NAS-Bench` for exploring and optimising architectures. The methods' hyperparameters are chosen to circumvent unfair advantages gained by increased compute time, e.g., no—iterations or function evaluations. Although we allocate similar resources for the baseline methods, it is difficult to reason for the fairness when comparing the baselines, when considering the disparity in the algorithmic approach of the baselines.

The bag-of-baselines implementation discussed in Izquierdo et al. (2021) are used and modified for compatibility with tabular benchmarks of `EC-NAS-Bench`. Each experiment is run for 10 trials using different initial seeds. All developed code will be made public upon the blind-review period ending.

**Random Search** The baseline methods, except for Random Search, apply evolutionary search heuristics to optimize architectures in the search space. The random search implementation samples architectures from the architecture uniformly at random, each time querying an architecture for a random epoch budget. Random search is done over 1000 iterations, as the other baseline methods, where applicable, will also run for 1000 iterations.

**Speeding up Evolutionary Multi-Objective Algorithm (SH-EMOA)** As with all our baselines, we use the implementation in Izquierdo et al. (2021). We define a problem for and search space following the bag-of-baselines API to allow model evaluation for different epoch budgets simply by querying the tabular benchmarks of `EC-NAS-Bench`. We initialize the algorithm with a population size of 250 and restrict the search to 1000 function evaluations for budgets between 4 and 108. However, we force the algorithm only to use budgets 4, 12, 36 and 108, which are available in our search space. The remaining hyperparameters we leave as default, which covers a uniform mutation type for architecture perturbation and tournament style parent selection for an off-spring generation.

**Mixed Surrogate Expected Hypervolume Improvement (MS-EHVI)** This evolutionary algorithm, too, is initialized with a population size of 250. We choose to generate 50 samples to lessen

---

**Procedure 3** LinearRankSample($P$, $\lambda$)

---

**Input:** set $P \subset \mathbb{X}$ of candidate solutions, number $\lambda$ of elements to be selected; reference point $\boldsymbol{r} \in \mathbb{R}^m$,
parameter controlling the preference for better ranked points $\eta^+ \in [1, 2]$
**Output:** $O \subset P, |O| = \lambda$

1: $O = \emptyset$
2: **for** $i \leftarrow 1$ to $m$ **do**
3: $\quad O \leftarrow O \cup \text{argmin}_{x \in P} f_i(x)$ $\qquad\qquad\qquad\qquad\qquad\qquad$ ▷ Always add extremes
4: **end for**
5: Compute $\Delta_{\boldsymbol{r}}(x, P)$ for all $x \in P$ $\qquad\qquad\qquad\qquad$ ▷ Compute contributing hypervolume
6: Sort $P$ according to $\Delta(x, P)$
7: Define discrete probability distribution $\pi$ over $P$ where

$$\pi_i = \frac{1}{|P|}\left(\eta^+ - 2(\eta^+ - 1)\frac{i-1}{|P|-1}\right)$$

$\quad$ is the probability of the element $x_i$ with the $i$th largest contributing hypervolume
8: **for** $i \leftarrow 1$ to $\lambda - m$ **do** $\qquad\qquad\qquad\qquad\qquad$ ▷ Randomly select remaining points
9: $\quad$ Draw $x \sim \pi$ $\qquad\qquad\qquad\qquad$ ▷ Select points with larger $\Delta_{\boldsymbol{r}}$ with higher probability
10: $\quad O \leftarrow O \cup x$
11: **end for**
12: **return** $O$

---

computation time, and we merely pass an auxiliary function to discretize parameters to fit with the experimental setup using tabular benchmarks.

**Simple Evolutionary Multi-Objective Algorithm (SEMOA)** Our MOO algorithm is described in subsection A.2. The key hyperparameters are the initial population size, which we set to 250, similar to the baseline methods, and likewise, we run the algorithm for 1000 iterations.

## B    MEASUREMENTS FROM CARBONTRACKER

We modify the open-source tool Carbontracker (Anthony et al., 2020) to measure the additional metrics reported in Table 1. Measurements take into account the energy usage of Graphical Processing Units (GPU), Central Processing Units (CPU), and Dynamic Random Access Memory (DRAM). Note that the energy usage for CPUs will include the power usage of DRAM. Power usage information is monitored, logged every 10 seconds, and reported as the average power usage during model training. Power is measured as the average of total units of a watt (W) over 10-second intervals during model training. The integral power consumed over the time a time interval, energy, is then reported in units of kilowatt-hours (kWh) with $1\text{kWh} = 3.6 \cdot 10^6$ Joule (J). Additionally, the emission of greenhouse gasses (GHG) is measured by equivalent units measured in grams of carbon dioxide ($CO_2$eq). The $CO_2$eq is then estimated by using the carbon intensity - $CO_2$eq units necessary to produce one unit of electricity a kilowatt per hour (kWh) - to express the carbon footprint of model training. The quantities for carbon intensity are fetched from the carbon intensity data provider every 15 minutes during model training.

Measurements from the aforementioned components alone do not give an accurate depiction of the carbon footprint model training when taking into account the energy consumption of the supporting infrastructure (e.g., data centre) is not considered. Therefore the quality of energy and carbon footprint estimations is amended by multiplying the power measurements by the PUE of the data centre hosting the compute resources. We use a PUE of 1.59, which is the global average for data centres in 2020 (Ascierto & Lawrence, 2020).

## C    ADDITIONAL RESULTS

The results in Figure 4 and Figure 5 were reported for the 5V- and 7V spaces respectively. The `EC-NAS-Bench` dataset also consists of the complete 4V space. In this section we report the MOO solutions based on the 4V search space. The trends observed for the 5V- and 7V spaces also hold for this smaller space as well.

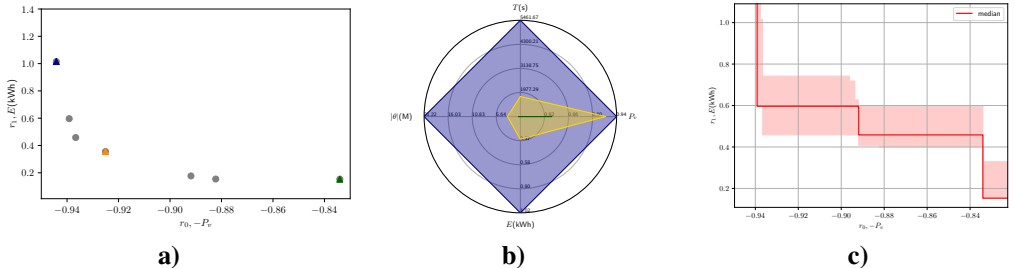

a)             b)             c)

Figure 6: Results for the 4V space. **a)** Pareto front obtained for one of the MOO runs shows the family of solutions as discrete points **b)** Radar plot showing three solutions from a Pareto front that is close to the median in subfigure a). The three architectures are chosen to correspond to the two extrema ($\mathcal{A}_{\mathbf{r}_0}$ in blue, $\mathcal{A}_{\mathbf{r}_1}$ in green) and the knee point ($\mathcal{A}_{\mathbf{r}_k}$ in yellow). The plot has four axes capturing the performance $P_v$ and resource consumption measured in $E, T, |\theta|$. **c)** The attainment curve showing the median solutions and the inter-quartile variations for 100 random initialisations of the MOO algorithm on the EC-NAS-Bench dataset. The two objectives being optimised $(-P_v, E)$ are shown on the two axes.

| Model | Strategy | $|\theta|$(M)↓ | $T(s)$ ↓ | $P_v$ ↑ | $E$(kWh)↓ |
|---|---|---|---|---|---|
| $\mathcal{A}_{\mathbf{r}_0}$ (B) | MOO | 21.22 | 5461.66 | 0.944 | 1.015 |
| $\mathcal{A}_{\mathbf{r}_1}$ (G) | MOO | **0.88** | **815.93** | 0.83 | **0.153** |
| $\mathcal{A}_{\mathbf{r}_k}$ (Y) | MOO | 3.20 | 1782.80 | 0.925 | 0.355 |
| $\mathcal{A}_{\mathbf{r}*}$ (R) | SOO | 21.22 | 5482.78 | **0.944** | 1.032 |

Table 4: Metrics for models in single- and multi-objective setting seen in Figure 6. For SOO the optimal solution ($\mathcal{A}_{\mathbf{r}*}$/Red) is reported. For MOO the two extrema ($\mathcal{A}_{\mathbf{r}_0}$/Blue, $\mathcal{A}_{\mathbf{r}_1}$/Green) and the knee point ($\mathcal{A}_{\mathbf{r}_k}$/yellow) are reported.

## D    SURROGATE ENERGY MODEL

The MLP-based surrogate model used to predict the training energy consumption of the 7v space, $E$ is given as: $f_\theta(\cdot) : \mathbf{x} \in \mathbb{R}^F \to E \in \mathbb{R}$, where $\theta$ are the trainable parameters and $\mathbf{x}$ comprises the features obtained from the architecture specifications. Using the cell/graph encoding of architectures introduced in Section 2.1, we populate $\mathbf{x}$ to consist of the upper triangular entries of the adjacency matrix, operations $\{$input, 1x1conv, 3x3conv, 3x3maxpool, output$\}$ mapped to categorical variables $[1, 2, 3, 4, 5]$, respectively and the total number of parameters. For the 7v space this results in $\mathbf{x} \in \mathbb{R}^{36}$.

We use a simple four layered MLP with gelu$(\cdot)$ activation functions, except for the final layer, which transforms the input in this sequence $36 \to 128 \to 64 \to 32 \to 1$.

The surrogate energy model is trained using actual energy measurements from 4300 randomly sampled architectures from the 7v space. The model was implemented in Pytorch (Paszke et al., 2019) and trained on an Nvidia RTX 3060 GPU. Using a training, validation and test split of ratio $[0.6, 0.1, 0.3]$ we train $f_\theta(\cdot)$ for 200 epochs with an initial learning rate of $5 \times 10^{-3}$ to minimise the the $L1$-norm loss function between the predicted and actual energy measurements using the Adam optimiser (Kingma & Ba, 2015).

