# OpenReview forum: "Energy Consumption-Aware Tabular Benchmarks for Neural Architecture Search"
_ICLR.cc/2023/Conference — Submitted to ICLR 2023_

### Official Review · Reviewer_gjRN · 2022-10-26

**Confidence:** 5
**Correctness:** 4
**Technical Novelty And Significance:** 2
**Empirical Novelty And Significance:** 1
**Recommendation:** 3

**Clarity, Quality, Novelty And Reproducibility:**

The paper is in general easy to follow. The authors provide the code with the supplementary material. The main issue with this submission is the novelty. When proposing a new benchmark, there should be some novel component in terms of the search space, benchmark construction (surrogate model used for instance in surrogate benchmarks), or empirical evaluations that provide major unforeseen insights into the field. Unfortunately, none of these criteria is fulfilled.

**Strength And Weaknesses:**

I find the proposed benchmark and the motivation for using such benchmark beneficial to the community as the awareness for resource efficient NAS increases. The paper is in general easy to follow and well-structured. The authors also release their codebase together with the API. However, I find the paper could improve with the following:

- **A larger and novel search space**. The smaller versions of the NAS-Bench-101 search space do seem way far from being realistic in my opinion. With 91 and 2532 architectures, respectively, they do not provide a realistic testbed and even the simplest search algorithms would lead to optimal solutions pretty quickly. Moreover, these spaces do not provide support for one-shot NAS methods.

- **More multi-objective algorithms to evaluate**. The authors only evaluate a single multi-objective algorithm in Section 4. It would be beneficial if they would add more methods to this section. Check [1] for some simple methods.

- **More on the benchmark than the algorithms used**. The authors spend most of Section 3 by describing the multi-objective optimization algorithm they use to evaluate on their benchmark. While such a detailed description is appreciated, it does not serve the main purpose of the paper and therefore I do not find it necessary to be included in the main paper. Rather, more statistics and analysis of the benchmark would be more useful.

-- References --

[1] Bag of Baselines for Multi-objective Joint Neural Architecture Search and Hyperparameter Optimization. Guerrero-Viu et al. 2021

**Summary Of The Paper:**

This paper proposes a tabular benchmark that includes various energy consumption metrics for a subset of networks in the NAS-Bench-101 search space. The authors provide simple multi-objective and single-objective baselines and run those on the benchmark.

**Summary Of The Review:**

The motivation for having a benchmark that provides energy consumption metrics is valid and would be beneficial for the NAS community. However, I think this submission is not ready for acceptance due to the reasons I mention above.

---

> ### Author Response · Authors · 2022-11-18
> **Response to reviewer**
>
> We thank the reviewer for their efforts and the constructive criticism of the work. We have tried to address the key concerns raised by them, and we hope this will be to their satisfaction.
>
> 1. **Large and novel search space**: We agree with the reviewer in that the experiments on the 4V and 5V spaces of NAS-Bench might not have been sufficiently representative. We refrained from experimenting with the 7V dataset due to the enormous resources required to populate them into the EC-NAS-Bench dataset. According to our estimates, it would have taken 770 GPU days, which was infeasible for us.
> Thanks to the reviewer's suggestion of a surrogate model, we have now implemented a surrogate energy prediction model. A simple four-layered MLP is used to predict the energy consumption of the 7V space based on the actual predictions from a random subset of 4300 architectures. The resulting surrogate model yields robust predictions (0.9977 Pearson correlation between predicted and actual energy predictions on the test set).
> Finally, we populate the EC-NAS-Bench to include the 7V surrogate space and perform MOO showing similar results as observed in the 4V and 5V spaces. That is, using energy consumption as an additional criterion yields inherently more efficient models.
> These points are also updated in the revised paper in Section 2.5 and in Figure 2. Details of the surrogate model are described in Appendix D.
> 2. **More MOO algorithms**: We again thank the reviewer for the useful pointer to [1]. We are embarrassed to acknowledge that we were not aware of this work, and hence was omitted in our initial submission.
> We have now used three of the methods from Bag-of-Baselines, primarily based on the ease of use and the time constraint (during the rebuttal period), along with the simple MOO algorithm we presented. Further, to sharpen the focus of the paper on the dataset, we have moved our MOO algorithm to Appendix A.
> Using attainment curves estimated from multiple runs of the MOO algorithms, we show that the Pareto-fronts obtained by all the methods are similar. This confirms the main hypothesis in this work, that adding energy consumption as a constraint accesses a sub-space of architectures that are inherently more efficient, and can be explored using any MOO algorithm.
> 3. **Additional insights on the dataset**: As mentioned above, we have now moved away from describing the MOO algorithm in the paper. Instead, we discuss the surrogate model for obtaining energy predictions in Section 2.5.
> Further, based on another reviewer's comments, we also study the influence of hardware on the energy measurements tabulated into EC-NAS-Bench. We run the 4V models on four different GPUs and observe the relative trends between models. And we show that, while the individual models may consume different energy on different GPUs, the overal trend is maintained across the models for a given hardware.
>
> Finally, we would like to point to the overall changes in this revised version which are also pointed out in the general comment on OpenReview. We have tried to address all the concerns raised by all reviewers, and it has substantially improved our paper. We sincerely hope the reviewer also recognises these modifications. We are willing to further clarify any other questions through the discussion period.
>
> [1] Bag of Baselines for Multi-objective Joint Neural Architecture Search and Hyperparameter Optimization. Guerrero-Viu et al. 2021

---

### Official Review · Reviewer_bAGs · 2022-10-29

**Confidence:** 3
**Correctness:** 3
**Technical Novelty And Significance:** 2
**Empirical Novelty And Significance:** 2
**Recommendation:** 5

**Clarity, Quality, Novelty And Reproducibility:**

Clarity: The paper is overall well-written. The author can improve the description of the MOO algorithm (e.g., give more details in the Appendix)

Quality + Novelty: The reviewer is concerned that it may be difficult for other researchers to continue their experiments on EC-NAS-Bench. The reason is that new hardwares appear every year and EC-NAS-Bench needs to be updated accordingly. The reviewer agrees that incorporting energy consumption in NAS is an important topic but EC-NAS-Bench won't be very useful when new hardwares appear.

Reproducibility: The author attached the source code. The reviewer has not ran the source code and only reviewed the content. It looks reproducible.

**Strength And Weaknesses:**

Strength

- Incorporating energy consumption data in NAS is an important problem. The author took the first step in this direction. Experiments show that it is able to find architecture that consumes much less energy but with comparable performance.

Weakness

- Different from the performance of the model, energy consumption of neural networks is a moving target. This is related to factors like global economy and the computational hardware. The author has not discussed how they will incorporate potential changes in the energy/power consumption and carbon footprint of the models.
- The multi-objective optimization algorithm is largely based on the cited paper "Multi-objective optimization with unbounded solution sets". Are there novel components proposed in the paper?

**Summary Of The Paper:**

The paper proposed an energy consumption-aware tabular benchmark for NAS based on NAS-101, called EC-NAS-Bench. EC-NAS-Bench contains the training energy consumption, power consumption and carbon footprint of each architecture in the benchmark. The author performs single-objective optimization and multi-object optimization (based on the algorithm proposed in the paper "Multi-objective optimization with unbounded solution sets") on EC-NAS-Bench, and noticed that multi-objective optimization is able to figure out architectures with about 70% energy reduction and <1% performance degradation


**Summary Of The Review:**

I vote for weak rejection because EC-NAS-Bench may soon outdated when new hardwares appear. Thus, this benchmark may not be useful to the research community.

---

> ### Author Response · Authors · 2022-11-18
> **Response to reviewer comments**
>
> We thank the reviewer for their effort and interest in our work. We address the key concerns here and also point to the relevant content in the revised paper.
>
> 1. **Influence of hardware**: The reviewer raises a very interesting point about the energy measurements being specific to the hardware we have trained on. We argue that while the specific energy consumption of the models might vary across hardware (such as GPUs), the relative trends between models should, largely, be maintained.
> To verify this claim, we trained the models in the 4V space on four different GPUs, spanning several generations. In Figure 3 in the revised paper, we show that although the exact energy consumed by a model is different between GPUs, the overall trends between models are maintained. This implies that when using the energy measurements from EC-NAS-Bench for MOO, the models that are preferred do not depend on the exact hardware they are trained upon.
> We do concede that some future developments in how GPUs are utilised for model training could favour a specific type of architectures more than others. One possible future work in this direction could be to train a surrogate model that can mimic different hardware configurations. Essentially, it would be possible to approximate the energy consumption of unseen hardware using surrogate models.
> The consistency in the trends and the specifications of the hardware experiments are detailed in Section 2.6 in the revised paper.
> 2. **MOO algorithms**: The MOO algorithm presented in the initial version of the paper was a simple method to demonstrate the usefulness of the search space with energy measurements. There were no significant novelties; and that was not the main contribution. As many reviewers have also pointed this out, we have moved our MOO algorithm description to Appendix A and incorporated three additional, existing MOO algorithms (Random search, SHEMOA and MSEHVI from [1]) in the revised paper.
>
> [1] Izquierdo, Sergio, et al. "Bag of baselines for multi-objective joint neural architecture search and hyperparameter optimisation." 8th ICML Workshop on Automated Machine Learning (AutoML). 2021.

---

> > ### Comment · Reviewer_bAGs · 2022-12-11
> > **Thanks for the rebuttal**
> >
> > I appreciate author’s response. However, I still think the paper lacks the novelty and it is difficult to maintain a reliable benchmark on energy consumption when there are multiple ways for selecting the hardware and calculating the energy consumption value.
> >
> > I thus choose to keep my score.

---

> > > ### Author Response · Authors · 2022-12-12
> > > **We thank the reviewer for the response and would like to comment.**
> > >
> > > We appreciate the reviewer’s response; however, we would like to refer to Figure 3 in Section 2.6 of the revised paper, which shows that the energy consumption of neural networks, albeit a moving target, behaves similarly for hardware selected across several generations of NVIDIA GPUs. Accordingly, we expect our benchmark to be useful for studying NAS algorithms in the future even if the hardware changes, because these changes will mostly be order-preserving w.r.t. energy consumption (people still use MNIST, GTSRB, … although they are based on outdated imaging technology 😉).

---

### Official Review · Reviewer_Bb6X · 2022-10-30

**Confidence:** 4
**Clarity, Quality, Novelty And Reproducibility:** 1. Clarity
**Correctness:** 3
**Technical Novelty And Significance:** 2
**Empirical Novelty And Significance:** 2
**Recommendation:** 5

**Strength And Weaknesses:**

**Strengths**: Energy consumption is indeed an important factor to consider in NAS. This benchmark can help facilitate further research and development on energy-efficient NAS. This work presents an interesting showcase of using MOO to find energy-efficient NAS.

**Questions and concerns**: This benchmark changed the search space of NAS-Bench-101. For example, instead of evaluating a 7V space, this work tests 5V and 4V spaces. Can the authors justify why this change is made? Clarity of the results: The results presented in Figure 2 are not very clear to me. Figure 2a shows the Pareto front obtained from one of the MOO runs. But what do the different markers and colors mean? It is my understanding that the blue, green, and yellow points correspond to the three types of architecture, including two extrema and a knee point. Why the red one, i.e., the optimal solution in SOO is, not shown in the figure? It would be great if the authors could establish benchmarked baseline performance on this new benchmark.

**Summary Of The Paper:**

This work proposes an energy consumption-aware tabular benchmark for NAS based on NAS-Bench-101. For each architecture, it adds the training energy consumption, power consumption, and carbon footprint. This work also demonstrates the usefulness of multi-objective architecture exploration for finding energy-efficient architectures without sacrificing much accuracy.
The MOO algorithm used is based on existing work from Krause et al. 2016.



**Summary Of The Review:**

The established energy-aware NAS benchmark is a very good contribution. However, the presentation of results could be improved. And benchmarked baselines should be added.

---

> ### Author Response · Authors · 2022-11-18
> **Response to reviewer comments**
>
> We thank the reviewer for their effort and interest in our work. The reviewer also raises several concerns, which we have addressed below:
> 1. **Search space size**: The initial search spaces used in this paper were only the 4V and 5V search spaces from the NAS-Bench101 dataset. The primary reason for not including the 7V search space was to reduce the energy consumption. Our estimations showed that to update the energy measurements as done for the 4V and 5V spaces to the 7V space would require about 770 GPU days. Further, we also argued that using these smaller spaces could sufficiently illustrate the points we were making. For instance, this is also an argument used in the NAS-Bench 101 paper [2], where the generalisation experiments are performed on a smaller dataset to conclude that the findings would extrapolate to larger search spaces.
> In the current version, however, we have extended the work to include the 7v search space comprising 423k architectures using a surrogate energy model, described in Section 2.5. We also report the MOO experiments on this larger dataset.
> 2. **Ambiguity of figures**: We apologise that the figures were not clear enough. As the reviewer points out, the markers depict the two extremal points and the knee point in the Pareto-front. The descriptions of these markers were embedded in the caption of Table 1, which we now also elaborate in the text in Section 4:
> >Metrics for models in single- and multi-objective settings are seen in Figure 4. For SOO the optimal solution ($\mathcal{A}\_{\mathbf{r}^*}$/Red) is reported. For MOO the two extrema ($\mathcal{A}\_{\mathbf{r}\_0}$/Blue, $\mathcal{A}\_{\mathbf{r}\_1}$/Green) and the knee point ($\mathcal{A}\_{\mathbf{r}\_k}$/yellow) are reported.
> 3. **Benchmark baselines**: We now report several existing MOO algorithms: random search, Speeding up Evolutionary Multi-Objective Algorithms (SHEMOA), and Mixed Surrogate Expected Hypervolume Improvement (MSEHVI), implemented in [1] to demonstrate the usefulness of the EC-NAS-Bench dataset.
>
> [1] Izquierdo, Sergio, et al. "Bag of baselines for multi-objective joint neural architecture search and hyperparameter optimisation." 8th ICML Workshop on Automated Machine Learning (AutoML). 2021.
> [2] Ying, Chris, et al. "Nas-bench-101: Towards reproducible neural architecture search." International Conference on Machine Learning. PMLR, 2019.

---

> > ### Comment · Reviewer_Bb6X · 2022-12-12
> > **Thanks for the response**
> >
> > I thank the authors for the response, which clarified the ambiguity in the figures. My concern about the search space remains. I prefer to keep my original rating for this work.

---

> > > ### Author Response · Authors · 2022-12-12
> > > **We thank the reviewer for the response and would like to comment.**
> > >
> > > We thank the reviewer for raising valid concerns about our work, which helped improving the quality and clarity of the paper. Considering the primary concern was the change to different search spaces between NAS-Bench-101 and EC-NAS-Bench, we would like to emphasize that our EC-NAS-Bench was extended to the entire 7V space during the rebuttal process, as described in Section 2.5, and in the revised version we empirically evaluate the 7V space as in NAS-Bench-101. In addition, we added baseline benchmarks for various MOO algorithms on EC-NAS-Bench.

---

### Official Review · Reviewer_FG4d · 2022-10-31

**Confidence:** 5
**Clarity, Quality, Novelty And Reproducibility:** 3) I'd like author clarify the follow…
**Correctness:** 3
**Technical Novelty And Significance:** 2
**Empirical Novelty And Significance:** 2
**Recommendation:** 5

**Strength And Weaknesses:**

Strength:
1) curate a new dataset and perform NAS on it.
2) the authors also evaluate their algorithms on NAS benchmark.
3) the paper is quite easy to follow, it is more like a tool paper that proposes a new benchmark for others to use.

Weakness:
1) I don't think it is necessary for such paper to introduce a new optimizer then spent a huge amount of content on the optimizer. Instead, I think it is more reasonable for authors to reuse the latest MOO algorithm such as [1] and [2] to perform NAS on their datasets, then release them as a baseline.

[1] Zhao, Yiyang, et al. "Multi-objective Optimization by Learning Space Partitions." arXiv preprint arXiv:2110.03173 (2021).

[2] Daulton, Samuel, Maximilian Balandat, and Eytan Bakshy. "Parallel bayesian optimization of multiple noisy objectives with expected hypervolume improvement." Advances in Neural Information Processing Systems 34 (2021): 2187-2200.

2) The experiments spent too much on the optimizer but not on the dataset, but I'd like to see more analysis on the dataset since I assume the focus of this paper is on introducing a new dataset.



**Summary Of The Paper:**

This paper present a new benchmark that extends NASBench-101 with the training energy footprints. The author also presents a MOO algorithm to perform the NAS.

**Summary Of The Review:**

It is a nice paper, but the authors seem to strike the reviews with some novelty. Frankly, it is okay for such tool paper to be not that novel as long as it solves practical problems. So I'd hope authors focus more on analyzing the dataset itself and establish a set of canonical baselines for using the dataset. Thank you.

---

> ### Author Response · Authors · 2022-11-18
> **Response to reviewer comments**
>
> We thank the reviewer for their constructive comments and the positive feedback. We have addressed the key concerns below. We hope it is to their satisfaction.
> 1. **MOO baselines**: We agree that the earlier version spent to much space on the MOO algorithm (which was meant to be a generic baseline). We have now moved our algorithm description to the Appendix A and introduced comparisons with three other baseline MOO methods in the main paper in Section 3.1 and Section 4. We have used the Bag-of-baselines methods [1] as suggested by one other reviewer, as this enabled us to set up multiple MOO algorithms within a single framework.
> 2. **More details on dataset**: In this updated version, we discuss the influence of energy measurements on different hardware in Section 2.6 and also introduce a surrogate energy prediction model for the larger 7V space in Section 2.5. The analysis of the surrogate model energy predictions and the number of training examples required are also presented in the updated Figure 2.
> 3. **Few-shot NAS**: The discussion of few-shot NAS is an interesting point when talking about energy efficiency, as they offer a reasonable trade-off between training from scratch and one-shot NAS. This is also pointed out by the reviewer. In our opinion, few-shot NAS could also benefit from energy consumption awareness if used along with a surrogate model. As we have demonstrated with the 7V space in the revised version of the paper in Section 2.5, reliable energy consumption metrics can be obtained when surrogate models are used. For the case of few-shot NAS, combining ideas from [2], which also predict the training accuracy, learning dynamics and our surrogate energy model, would be one possible solution.
> We now cite relevant literature for few-shot NAS pointed out by the reviewer, and add a discussion point about few-shot NAS in Section 5.
>
> [1] Bag of Baselines for Multi-objective Joint Neural Architecture Search and Hyperparameter Optimization. Guerrero-Viu et al. 2021
> [2] Zela, Arber, et al. "Surrogate NAS benchmarks: Going beyond the limited search spaces of tabular NAS benchmarks." Tenth International Conference on Learning Representations. 2022.

---

### Author Response · Authors · 2022-11-18
**Summary of rebuttal addressed to all reviewers and the area chair**

We thank all four reviewers for their effort and constructive feedback on our work. While most reviewers seemed to be largely positive about the scope and usefulness of our work, they have also raised multiple concerns, which, however, we have addressed in the revised version.

Taking the feedback of all four reviewers into consideration, we have substantially updated the paper in the following ways:

1. **Focus on dataset**: We have moved the description of the multi-objective optimisation (MOO) algorithm introduced in this work to Appendix A, to sharpen the focus of the main paper on the EC-NAS-Bench dataset.
2. **Comparison with other MOO algorithms**: Our goal was not to promote any MOO algorithm, but to provide a proof of concept. However, we agree with the reviewers in that introducing a simple MOO algorithm without comparing it with existing MOO methods was a weakness. We have addressed this by comparing our simple evolutionary MOO algorithm (SEMOA) with three other MOO algorithms from [1]. The specific choice of the three MOO algorithms from [1] (Random search, SHEMOA and MSIHVE) was primarily due to the time constraint and ease of use.
3. **Experiments on a larger dataset**: Keeping the main goal of this work in sight -- to minimise energy consumption -- we restricted our initial experiments to the 4V- and 5V spaces. We recognise this is a weakness but performing experiments on the 7V space was prohibitively expensive (770 GPU days). Taking the feedback from the reviewers into consideration, we now introduce a surrogate EC-NAS-Bench dataset for the 7V space, similar to recent surrogate NAS methods like [2]. We train a multi-layered perceptron (MLP) on a subset of architectures from the 7V space with actual energy consumption and use this surrogate model to predict on the remaining 7V space. We then use this surrogate 7V dataset in our MOO experiments. We study the robustness of these surrogate predictions in Fig. 2 and find it to be sufficiently reliable (Pearson correlation between predicted and actual energy measurements = 0.9977) in the revised version of the paper.
4. **Influence of hardware**: One of the reviewers raised an important concern about the generalisation of the reported energy measurements, which are specific to a single GPU we used. We argue that the trends of energy consumption of all models are maintained irrespective of the GPU used. To confirm this claim, we now measure the energy consumption of the 4V space on four different GPUs, and show that the relative energy consumption between models is maintained across hardware. This is discussed in Section 2.6.
5. **Other minor changes**: Due to the inclusion of the MOO algorithms, we have reduced the focus on comparisons with the single objective optimisation (SOO) methods. We have also removed the complex radar plot for SOO to include new figures.

In summary, we have incorporated all the major criticisms from all the reviewers resulting in an improved version of the paper. We have now shown that jointly optimising the energy consumption and performance when performing NAS yields efficient architectures; this is true irrespective of the search space (4V, 5V, 7V), MOO algorithms used, and the hardware the models are trained on. We sincerely hope that the reviewers and Area Chair recognise these improvements. We are open to further clarifying the revisions or addressing any further comments through the discussion period.

**Note**: The updated paper now has the new text and figure (captions) in blue and the content that has been moved from the main text to Appendix in red.

#### References:
[1] Bag of Baselines for Multi-objective Joint Neural Architecture Search and Hyperparameter Optimization. Guerrero-Viu et al. 2021
[2] Zela, Arber, et al. "Surrogate NAS benchmarks: Going beyond the limited search spaces of tabular NAS benchmarks." Tenth International Conference on Learning Representations. 2022.

---

### Comment · Area_Chair_AH6D · 2022-12-07
**Response to Author Feedback**

Dear Reviewers, thank you so much again for your time on this paper. The discussion phase is still ongoing, how does the author response and other reviews change your view of the paper?

---

### Decision · Program_Chairs · 2023-01-20

**Decision:**

Reject

**Justification For Why Not Higher Score:**

See above metareview.

**Justification For Why Not Lower Score:**

N/A

**Metareview: Summary, Strengths And Weaknesses:**

While the reviewers appreciated the paper’s motivation, the idea of using MOO for energy-efficient NAS, and the clarity of the paper, their main concerns were with (a) the realism of the benchmark, and (b) the divided focus of the paper. Specifically, for (a) they argued that smaller versions of the NAS-Bench-101 search space seem too small (91, 2532 architectures) to provide a realistic testbed for NAS algorithms. They argue that simple search algorithms which would work well here, may not work well in a more realistic setting with tens or hundreds of thousands of possible architectures. The authors responded by agreeing that those datasets are not sufficiently representative. They point out the computational complexity of obtaining results for their larger proposed benchmark, but then say they take a tip from the reviewer about using a surrogate model to predict energy consumption for this larger benchmark. To train this surrogate they subsample 4300 architectures (out of 423,000) and learn an MLP to predict energy consumption. They report that this model is accurate (0.9977 Pearson correlation). This speeds up training significantly and they present new results with this surrogate model. However, as far as I can tell, surrogate accuracy is computed with respect to the same subsampled architectures that were used to train the surrogate model. So it’s not at all surprising that they were able to achieve such a high correlation with an MLP. There is no indication that this surrogate is accurate for architectures outside of the training set. For this reason, the issue of a useable realistic energy-aware NAS benchmark is still open. For (b), the reviewers were confused that the paper seemed to spend so much space presenting a novel MOO method when the main goal of the paper seemed to be presenting a benchmark. The authors responded by shrinking this section, using the space to describe the surrogate above. I think this was an improvement, but there is still 3/4th of a page free in the paper, it’s unclear why say, Appendix C is not moved to the main paper given the additional space. For these reasons I vote to reject at this time. Once these things are fixed this paper will be improved.